# Investigating Self-Attention: Its Impact on Sample Efficiency in Deep Reinforcement Learning

## Abstract

Improving the sample efficiency of deep reinforcement learning (DRL) agents has been an ongoing challenge in research and real-world applications. Self-attention, a mechanism originally popularized in natural language processing, has shown great potential in enhancing sample efficiency when integrated with traditional DRL algorithms. However, the impact of self-attention mechanisms on the sample efficiency of DRL models has not been fully studied. In this paper, we ponder the fundamental operation of the self-attention mechanism in visual-based DRL settings and systematically investigate how different types of scaled dot-product attention affect the sample efficiency of the DRL algorithms. We design and evaluate the performance of our self-attention DRL models in the Arcade Learning Environment. Our results suggest that each self-attention module design has a distinct impact on the sample complexity of the DRL agent. To understand the influence of self-attention modules on the learning process, we conduct an interpretability study focusing on state representation and exploration. From our initial findings, the interplay between feature extraction, action selection, and reward collection is influenced subtly by the inductive biases of the proposed self-attention modules. This work contributes to the ongoing efforts to optimize DRL architectures, offering insights into the mechanisms that can enhance their performance in data-scarce scenarios.

## 1 Introduction

Deep reinforcement learning (DRL) (Arulkumaran et al., 2017; Li, 2017; François-Lavet et al., 2018) is a branch of machine learning (Jordan & Mitchell, 2015) that combines the art of decision-making of reinforcement learning (RL) (Sutton, 2018) with the representation learning capabilities of deep neural networks (Goodfellow et al., 2016). It has made tremendous progress in advancing AI for its ability to solve complex sequential decision-making problems that traditional algorithms struggle with, especially in environments with high-dimensional sensory inputs and where the optimal solutions are unknown or difficult to model explicitly. This makes DRL particularly well-suited for tasks like robotics (Morales et al., 2021; Ibarz et al., 2021), game-playing (Mnih, 2013; Mnih et al., 2015; Silver et al., 2018; Vinyals et al., 2019), autonomous vehicles (Kiran et al., 2021), healthcare (Yu et al., 2021), and financial markets (Hu & Lin, 2019; Hambly et al., 2023), where agents must learn and adapt from experience without being directly programmed with rules. However, one of the key challenges in DRL is sample inefficiency (Yu, 2018; Yarats et al., 2021). DRL algorithms often require an extensive amount of interactions with the environment to learn effectively, making them computationally expensive and time-consuming. In real-world applications, especially where data collection is costly or time-constrained (e.g., robotics or medical treatments), this inefficiency becomes a major bottleneck. Addressing the sample inefficiency issue is crucial to improving the practicality and scalability of DRL algorithms, driving research into methods that can enhance learning efficiency with fewer training samples.

State-of-the-art approaches to improve sample efficiency in DRL focus on several strategies, including sample reuse such as experience replay (Fedus et al., 2020) and prioritized experience replay (Schaul, 2015), model-based reinforcement learning (Kaiser et al., 2019; Schrittwieser et al., 2020;

Hafner et al., 2020; Schwarzer et al., 2020; Micheli et al., 2022; Kapturowski et al., 2022; Moerland et al., 2023), transfer learning (Spector & Belongie, 2018; Yang et al., 2021; Liu et al., 2021; Zhu et al., 2023), meta-learning (Sung et al., 2017; Liu et al., 2019; Rakelly et al., 2019; Franke et al., 2020; Beck et al., 2023), and leveraging advanced neural network architectures (Chen et al., 2021; Schwarzer et al., 2023). Recently, self-attention mechanisms (Vaswani, 2017), commonly used in natural language processing (like in Transformers (Han et al., 2022)), have been applied to DRL to enhance sample efficiency (Manchin et al., 2019; Shen et al., 2019; Hu et al., 2019; Chen et al., 2020; Fernandes et al., 2023). While most of the existing approaches focus on integration techniques of self-attention in DRL, we ponder the fundamental question of how the scaled dot-product attention proposed in the original Transformer (Vaswani, 2017) can be optimally devised for visual-based DRL tasks. Specifically, we take a closer look at how applying dot product over different dimensions of the query, key, and value tensors affects the sample efficiency of the DRL algorithms.

In this work, we focus on investigating the underlying operation of the self-attention mechanism and its impact on sample efficiency by designing various self-attention modules and evaluating them with a baseline RL algorithm in the Arcade Learning Environment (ALE) (Bellemare et al., 2013; Machado et al., 2018). Our results indicate that each self-attention module influences the agent's learning process differently, driven by its unique inductive bias (Baxter, 2000; Utgoff, 2012; Goyal & Bengio, 2022). Furthermore, we perform an interpretability study to provide better insights into how various self-attention modules influence sample efficiency through the lens of state representation and exploration. Our initial observations suggest that self-attention modules can introduce artifacts that subtly impact the agent's learning process. We picture the proposed self-attention modules in Section 4, illustrate the experiment setup, and present the main results in Section 5.

## 2 RELATED WORK

In the field of DRL, improving sample efficiency has been a critical research focus. Most recent works try to tackle the sample efficiency challenge via model-based RL (Hafner et al., 2020; Schwarzer et al., 2020; Micheli et al., 2022; Kapturowski et al., 2022) where agents build a model of the environment to simulate interactions, reducing the need for actual interactions. Although model-based RL has made significant progress in lowering the sample complexity, it comes with notable limitations such as model inaccuracy, high computational cost, and limited applicability in real-world environments (Doll et al., 2012; Clavera et al., 2018; Pong et al., 2018).

Considering the limited literature on improving sample efficiency through self-attention, we focus on the most relevant research related to our study in this section. In Manchin et al. (2019)'s work, self-attention has been integrated with the Proximal Policy Optimization (PPO) algorithm (Schulman et al., 2017) to address the sample complexity issue and has shown great potential in setting new state-of-the-art results in the ALE benchmark. Specifically in the context of ALE, the input sequence is a stack of images, and the query, key, and value are generated by applying a $1 \times 1$ Convolutional Neural Network (CNN) kernel over the feature maps of the first CNN layer. The scaled dot-product attention is then computed between the query, key, and value to generate the attention maps. The attention maps are then element-wise summed with the feature maps from the first CNN layer before being passed to the second CNN layer. The work further explored various ways of integrating the self-attention block and evaluated their performances over 40 million time steps across 10 games using 3 random seeds. Our work differs from it in multiple perspectives. Firstly, we focus on the fundamental operation of self-attention in terms of the dimensions where the scaled dot-product should be applied. Secondly, we evaluate our proposed self-attention agents over 10 million time steps across 56 games with 5 runs per game. Thirdly, we provide insights into how self-attention influences the agent's learning process in terms of state representation and exploration with the consideration of the inherent inductive biases of the self-attention modules.

## 3 PRELIMINARIES

### 3.1 PPO

Proximal Policy Optimization (PPO) (Schulman et al., 2017) is a model-free, policy gradient RL algorithm that has become the de facto choice for many RL tasks due to its data efficiency, reliability,

and scalability. It builds upon the TRPO algorithm (Schulman, 2015) with the key improvement of enabling multiple epochs of minibatch updates. PPO is typically implemented using an actor-critic framework where the actor is the policy network that selects actions and the critic is the value network that estimates the value of a state or state-action pair. The critic helps guide the actor by providing more accurate value estimates, improving learning efficiency. To encourage exploration, PPO often includes an entropy term in the objective. Higher entropy indicates more randomness in the agent's action selection, which can prevent premature convergence to suboptimal policies. Considering PPO's general advantages over other RL algorithms, we choose PPO as the baseline agent for evaluating the performance of our proposed self-attention modules.

## 3.2 SELF-ATTENTION

The self-attention mechanism is a core component in many modern neural networks, particularly in architectures like the Transformer (Vaswani, 2017), and it is widely used in tasks such as natural language processing (NLP), computer vision, and more recently, reinforcement learning. Self-attention (a.k.a. intra-attention) refers to the mechanism where a sequence element attends to other elements within the same sequence. It computes relationships between all pairs of elements in the sequence, allowing the model to capture dependencies regardless of the distance between them. Specifically, the self-attention proposed in the Transformer is termed the scaled dot-product attention which is the primary focus of this paper. We outline the mathematical formulation of the scaled dot-product attention in the context of NLP as follows.

- For each input element in a sequence, generate the query, key, and value vectors as $q, k, v$ with the key vector having the dimension of $d_k$

- For the entire sequence, pack all the queries, keys, and values into matrices as $Q, K, V$

- Compute the attention scores matrix as $Attention(Q, K, V) = softmax(\frac{QK^T}{\sqrt{d_k}})V$

It is important to note the differences between the self-attention formulation in the general NLP settings (like the one defined here) and the self-attention formulation proposed by Manchin et al. (2019). The fundamental difference lies in how query, key, and value are generated and those 3 components are no longer represented as matrices but as 3D tensors in the latter case. This key change in the representation of the query, key, and value catalyzes the core direction of our research.

## 4 DESIGN OF SELF-ATTENTION MODULES

Inspired by Manchin et al. (2019)'s work, we embark on a study to explore the impact of various forms of scaled dot-product attention on the sample efficiency of the PPO algorithm. As depicted in Figure 1, each self-attention module encircled by the dashed line is positioned between the first and the second CNN layers within the state representation block (a.k.a. feature extractor) of the PPO framework. The key reason for placing the self-attention module at such a location is to enhance computational efficiency and preserve interpretability. To shed more light on this, positioning the self-attention module before the first CNN layer would result in higher computational costs due to the high dimensionality of raw observations, and placing it after the second CNN layer would reduce interpretability, as features become more abstract at this stage.

Zooming into each self-attention module, the query **Q**, key **K**, and value **V** tensors are generated individually by applying a $1 \times 1$ CNN kernel over the feature maps $\mathbf{F}_1$ at the first CNN layer H1. As a result, the dimensions of **Q**, **K**, and **V** match those of $\mathbf{F}_1$, including the channel, row (height), and column (width) dimensions. We vary the design of self-attention modules by permuting the dimensions of the query, key, and value tensors, allowing the scaled dot-product attention to be applied across different dimensions. We designate the proposed self-attention modules according to the dimensions over which the dot product is performed. For simplicity, we omit the term "self" from "self-attention" for all modules.

- **S**patial-**w**ise-**A**ttention (**SWA**): dot product is applied over the row and column dimensions and repeated along the channel dimension

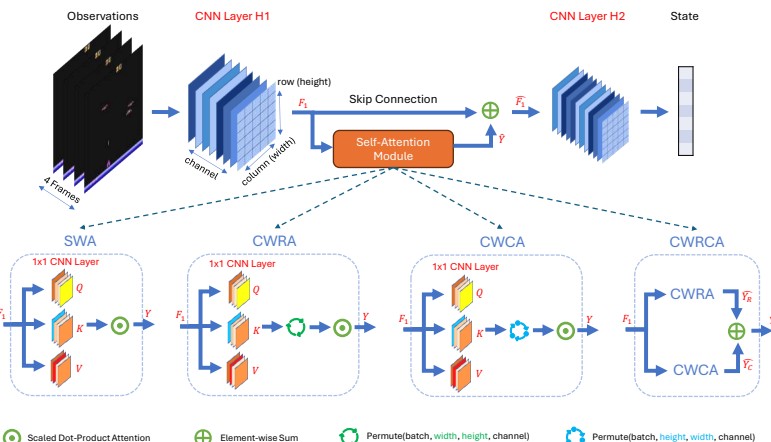

Figure 1: **Design of self-attention modules**. Each self-attention module enclosed by the dashed lines is placed between the first and the second CNN layers within the state representation block of the PPO's network architecture. The query $\mathbf{Q}$, key $\mathbf{K}$, and value $\mathbf{V}$ tensors are generated individually by applying a $1 \times 1$ CNN kernel over the feature maps $\mathbf{F}_1$ produced at the first CNN layer H1. Specifically in CWRA and CWCA, the order of the dimensions of the query, key, and value tensors is permuted such that the scaled dot-product attention is applied over different dimensions. The outputs of SWA, CWRA, and CWCA labeled as $\mathbf{Y}$ are subsequently reshaped into $\hat{\mathbf{Y}}$ which has the same shape as the feature maps $\mathbf{F}_1$ before the element-wise summation. Particularly in CWRCA, the outputs of CWRA and CWCA are reshaped into $\hat{\mathbf{Y}}_R$ and $\hat{\mathbf{Y}}_C$ respectively (both having the same shape as $\mathbf{F}_1$) before being summed. The attended feature maps $\hat{\mathbf{F}}_1$ produced by the sum of $\mathbf{F}_1$ and $\hat{\mathbf{Y}}$ are then passed to the second CNN layer H2 for state representation. The complete network architecture of PPO is presented in Appendix A.

- **C**hannel-**w**ise-**R**ow-**A**ttention (**CWRA**): dot product is applied over the channel and row dimensions and repeated along the column dimension

- **C**hannel-**w**ise-**C**olumn-**A**ttention (**CWCA**): dot product is applied over the channel and column dimensions and repeated along the row dimension

- **C**hannel-**w**ise-**R**ow-**C**olumn-**A**ttention (**CWRCA**): this is simply the element-wise sum of the outputs of **CWRA** and **CWCA**

Note that the permutation operation is done before and after the scaled dot-product attention in CWRA and CWCA modules where the post-permutation (i.e., reshaping of $\mathbf{Y}$ into $\hat{\mathbf{Y}}$) ensures that the final attention maps $\hat{\mathbf{Y}}$ are compatible with the feature maps $\mathbf{F}_1$ before the element-wise summation. For ease of comparison, we denote the baseline PPO algorithm as **NA** which stands for **N**o-**A**ttention and illustrate its architecture in Appendix A. We argue that each self-attention module has its own inductive bias when integrated with the baseline algorithm and plays a distinct role in the RL feedback loop. In the context of ALE, every environment has its unique game mechanics. We believe that whether the inductive bias of the self-attention module would enhance or impair learning is highly dependent on the game mechanics of the environment. For example, the CWRA module assumes attending features lie in the channel, and the row dimensions could benefit the agent's learning. In other words, agents equipped with the CWRA module could learn faster in games with rewarding objects moving along the column dimension, i.e., larger variance or higher degree of dynamics exist in the horizontal direction of the game screen. On the contrary, agents equipped with the CWCA module may learn faster in games with rewarding objects moving along the row dimension, i.e., larger variance or higher degree of dynamics present in the vertical direction of the game screen. We highlight the observations that generally support this belief in Section 5.

## 5 EXPERIMENT

This section documents the experiment's setup, presents the main results, and discusses the key findings.

### 5.1 EXPERIMENT SETUP

To assess the impact of the proposed self-attention modules on sample efficiency, we compare the performance of self-attention-enabled agents with that of a baseline agent using the well-established ALE benchmark. Specifically, each agent is trained for 10 million time steps across 56 games with 5 random seeds using the RL Baselines3 Zoo v2.0.0 (Raffin, 2020; Raffin et al., 2021) training framework. We detail the hyperparameters used in Appendix B for reproducibility.

### 5.2 RESULTS AND ANALYSIS

**Evaluation Methodology** We follow the best practices recommended by Agarwal et al. (2021) for reliable evaluation of the agent's performance. In particular, we report performances with 95% stratified bootstrap confidence intervals (CIs) based on human normalized scores (HNS). To compute HNS, we obtain the performance of a random agent $score\_random$ and the performance of an averaged human player $score\_human$ from Badia et al. (2020) and normalize the performance of our agents $score\_agent$ using $HNS = \dfrac{score\_agent - score\_random}{score\_human - score\_random}$. In addition, we use the mean evaluation score over the entire evaluation period instead of the last evaluation score for score normalization and stratified bootstrapping. The reason is twofold: 1) The mean evaluation score over the entire evaluation period favors sample efficiency whereas the last evaluation score favors the final performance; 2) Using the mean evaluation score over the entire evaluation period for stratified bootstrapping generally results in smaller CIs than those generated using the last evaluation score. Since all agents share the same underlying algorithm (PPO) and differ only in their feature extractors (particularly the self-attention modules), we anticipate minor performance variations due to the stratified bootstrapping process. Backed by the no-free-lunch theorem (Wolpert et al., 1995; Wolpert & Macready, 1997; Baxter, 2000), we present evaluation results using the sample mean and standard error per game to better illustrate the impact of each self-attention module's inductive bias within specific game environments. To reduce statistical uncertainty, the mean evaluation score over the entire evaluation period is used to calculate the sample mean and standard error.

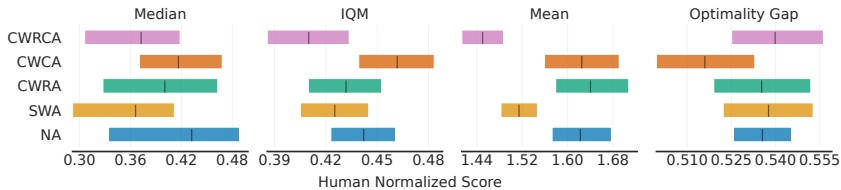

Figure 2: **Aggregate performance** (Agarwal et al., 2021). The median, interquartile mean (IQM), mean, and optimality gap based on human normalized scores (HNS) of each agent are shown with 95% stratified bootstrap confidence intervals (CIs) from left to right. We focus on the IQM and optimality gap in favor of their robustness and statistical efficiency. Although most CIs overlap under each metric, the CWCA agent achieved a higher IQM score and a lower optimality gap, and the CWRCA agent showed lower overall performance across all metrics.

**Overall Performance with Stratified Bootstrap CIs** The aggregate performance, sample efficiency curves, and performance profiles of all the agents are depicted in Figure 2, 3a, and 3b respectively. The average probability of improvement between any of the two agents can be found in Appendix C. In general, we observe relatively small differences in agents' performances, likely due to the shared baseline algorithm used across designs. This implies that the proposed self-attention models and the baseline model have similar overall performance when all 56 games are considered. Nevertheless, the CWCA agent exhibits slightly better performance in terms of IQM (the higher the better), optimality gap (the smaller the better), and sample efficiency (the higher the better) whereas the CWRCA agent demonstrates relatively inferior performance in terms of all evaluation metrics.

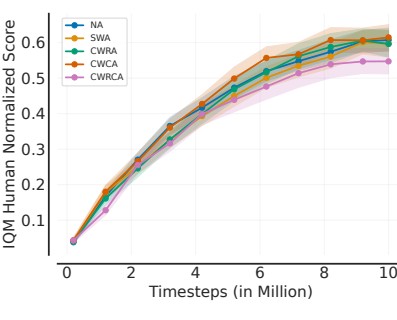 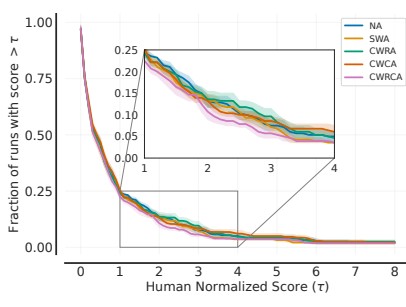

(a) Sample efficiency curves        (b) Performance profiles

Figure 3: **Sample efficiency curves (left) and performance profiles (right)** (Agarwal et al., 2021). Sample efficiency of the agents is represented using IQM human normalized scores at selected time steps over the entire evaluation period. Shaded regions show point-wise 95% stratified bootstrap CIs. The CWCA agent demonstrated slightly higher sample efficiency, consistent with its IQM performance as presented in Figure 2. The performance profiles are plotted based on score distributions. Although the score distributions of all agents look similar at first glance, the profiles intersect at multiple points where $\tau \in [1, 4]$ which implies that there is no stochastic dominance among all agents. In other words, each agent could perform differently in different games.

Another observation from the performance profiles highlights the performance delta among agents over a specific range of HNS thresholds (i.e., $\tau \in [1, 4]$). We argue that although the stratified bootstrapping process aims to provide more reliable evaluation results by accounting for uncertainty in the few-run regime (Agarwal et al., 2021), it could fade the manifestation of agents' unique characteristics in specific games, such as their inductive biases. This is likely true in our context where all agents share the same underlying learning mechanism, i.e., the PPO backbone.

**Inductive Biases and Game Mechanics**     In the pursuit of discovering the effect of inductive biases of the proposed self-attention modules, we include the performance of the agent per game in Appendix D. For each game, we compute the sample mean and standard error using the mean evaluation score over the entire evaluation period across 5 runs. Based on the highest sample mean, we select the winning agent per game and summarize the list of games won by each agent. Since there is no quantitative way to measure the game mechanics and the inductive biases of the self-attention modules, we intend to correlate these two concepts in an empirical and heuristic manner. For each self-attention-enabled agent, we choose the game where the agent exhibits a relatively higher winning margin in terms of the sample mean and a relatively lower standard error as the representative game to study the relationship between the inductive bias of the self-attention module and the game mechanics. We present the list of the representatives in Figure 4 and the complete list of games won by each agent in Appendix E.

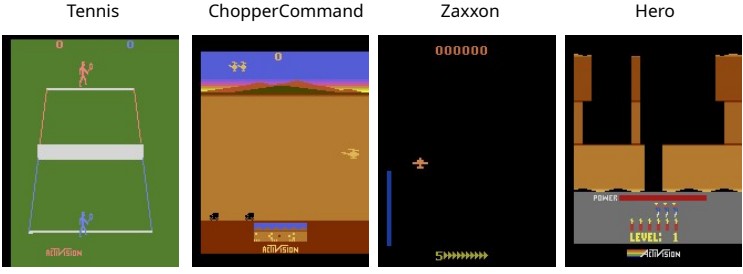

Figure 4: **Inductive biases and game mechanics**. The representative games selected for the SWA, CWRA, CWCA, and CWRCA agents are shown from left to right.

Tennis can be considered as a fully observable game (Kapturowski et al., 2022) with most of the features and dynamics available in the spatial domain, i.e., the row and column dimensions, and has a relatively static background (lower dynamics in the channel dimension). We hypothesize that the

SWA module could 'exploit' its inductive bias more naturally in the Tennis environment to obtain higher rewards. ChopperCommand is a horizontally scrolling shooter. We observe that both the chopper and the targets are mostly moving horizontally which could be 'leveraged' by the CWRA module with its inherent attention over the channel and row dimensions. In contrast to ChopperCommand, Zaxxon is a vertically scrolling shooter. The vertical movements of the spaceship, targets, and fortresses could be 'taken advantage of' by the inductive bias of the CWCA module. As for the CWRCA module, intuitively, it could 'combine' the strengths of both CWRA and CWCA modules. Albeit having the most complex design, it excels in the game of Hero where the rescuer traverses down a mineshaft avoiding enemies and hazards, and destroying walls to rescue trapped miners. Heuristically, Hero is a highly exploratory game that demands attention or curiosity in all directions. We conjecture that the CWRCA module could encourage exploration by creating state representation with high entropy (Vuckovic et al., 2020; Zhao et al., 2021). We show some observations that could underpin this hypothesis in Section 5.3.

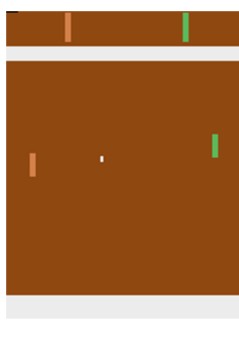

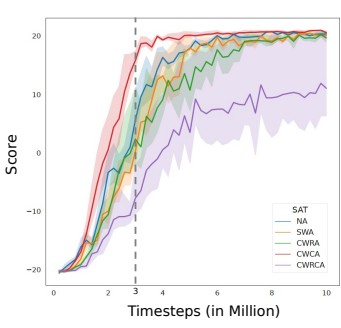

(a) The Pong game

(b) Learning curves in Pong

Figure 5: **The Pong game (left) and the learning curves in Pong (right)**. For learning curves, solid lines represent the mean performance, and the shaded regions indicate the 95% confidence intervals across 5 runs. Model checkpoints are selected at the 3 million time step for interpretability study.

## 5.3 INTERPRETABILITY STUDY

To further understand how self-attention modules influence the sample efficiency of the baseline algorithm, we suggest interpreting the inner workings of the self-attention mechanisms from the perspectives of state representation and exploration. Without loss of generality, we selected the Pong game for our initial case study for the following reasons.

- Pong has a simple state space where the game features two paddles (left and right), a ball, and two walls (top and bottom) as shown in Figure 5a. The Pong game simulates table tennis where the left paddle is manipulated by the game emulator and the right paddle is controlled by a learning agent. Having a simple state space, features or artifacts created by the self-attention modules could be spotted easily.

- Pong also has a relatively small action space with a total of 6 default actions, namely, 'NOOP' (no operation, do nothing), 'FIRE', 'RIGHT' (move the paddle up), 'LEFT' (move the paddle down), 'RIGHTFIRE', and 'LEFTFIRE'. Specifically in Pong, 'FIRE' has the same effect as 'NOOP', 'RIGHT' is equivalent to 'RIGHTFIRE', and 'LEFT' is equivalent to 'LEFTFIRE'. This further reduces the action space to 3 distinctive actions which could ease our analysis.

- In addition, the learning curves depicted in Figure 5b exhibit a clear separation among agents, especially between the CWCA and the rest of the agents. We believe that the more distinguishable the learning curves, the larger the distinction in agents' state representations and behaviors. Under this assumption, we select the model checkpoint at the 3 million time step (indicated by the dashed line) where we observe a relatively large variation in agents' performances. For each agent, we pick the saved model from a specific run whose learning curve resembles its mean performance the most and use that saved model to reproduce the agent's behaviors at that particular checkpoint. We summarize the learning curves of all games in Appendix F.

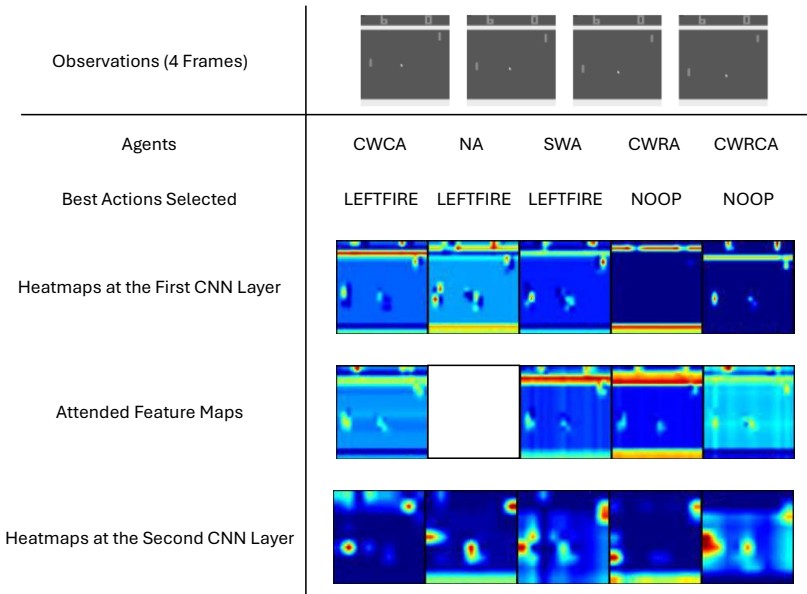

Figure 6: **State representation**. The heatmaps are generated using Grad-CAM (Selvaraju et al., 2016) and the attended feature maps are generated via element-wise summation between the attention maps and the feature maps at the first CNN layer. The attended feature map of the NA agent (i.e., the baseline agent) is blank because it does not contain any self-attention module. The artifacts created by the self-attention modules are most noticeable in the attended feature maps of the CWRCA agent where horizontal and vertical bars can be observed. We hypothesize that these artifacts could slow down the learning process of the CWRCA agent because the agent needs to learn to disentangle patterns that are naturally present in the game scene and created by the self-attention modules. The heatmaps and the attended feature maps of all observations are provided in Appendix G.

**State Representation**    To visualize the state representation and its correlation with the agent's behavior, we make use of the Grad-CAM (Selvaraju et al., 2016) to generate the gradient-weighted feature maps (denoted as heatmaps) at the first and the second CNN layers where the gradient is back-propagated from the logit of the 'best' action selected by the agent. To visualize the effect of self-attention modules in the feature space, we extract the attended feature maps $\hat{\mathbf{F}}_1$ (as indicated in Figure 1) which are the element-wise sum of the attention maps $\hat{\mathbf{Y}}$ and the feature maps $\mathbf{F}_1$ generated at the first CNN layer. To better understand the feature extraction process, 10 unique sets of observations are selected using a random policy and fed into the trained agents to retrieve the heatmaps and the attended feature maps. In favor of simplicity, we pick one set of observations and organize corresponding heatmaps and attended feature maps from each agent in Figure 6. The observations comprise four consecutive frames that depict the ball's movement from the center to the bottom right of the screen where the agent is located at the top right corner. In this situation, a human player would start to move the paddle down, i.e., pick the action 'LEFT' or 'LEFTFIRE' to catch the ball to avoid losing the score. Looking at the heatmaps at the first CNN layer and the best action selected by each agent, all agents could highlight the state information including the ball, the paddles, and the walls except for the CWRA agent which focused more on the walls. The difference in the heatmap is likely caused by the less optimal action chosen by the CWRA agent in the sense that 'doing nothing' at this moment will probably lead to a score loss later. It is counter-intuitive that the CWRCA agent chose the same action 'NOOP' but with the 'right' state representation. This implies that the CWRCA agent may not have fully grasped the correct correlation between the optimal action and the current state.

Based on the attended feature maps, the SWA and the CWRA agents focused more on the walls whereas the CWCA agent paid equal attention to all the key objects in the scene. An interesting phenomenon observed is the creation of artifacts (patterns not naturally present in the game scene)

from the self-attention modules. Different self-attention modules seem to create different patterns of artifacts according to their dot product operations. The artifacts created by the CWCA module resemble a horizontal bar (like walls) whereas artifacts created by the SWA module resemble multiple vertical bars (like 'transposed' walls). Likewise, artifacts created by the CWRCA module contain both horizontal and vertical patterns. We notice that such artifacts seem to be present in the heatmaps of the second CNN layer as well, especially for the SWA and CWRCA models. It is shown that self-attention modules are capable of creating various patterns of artifacts based on the game scenes and the artifacts could behave like a double-edged sword in the sense that they can influence an agent's learning both positively (e.g., when artifacts overlay with the actual features) and negatively (e.g., when artifacts ambiguate the state representation). In the context of Pong, vertical artifacts seem to do more harm to the learning likely because the model has to learn to disambiguate between the artifacts and the actual paddles and walls. The negative impact of these vertical artifacts is manifested by the slower learning curves of both SWA and CWRCA models as shown in Figure 5b. Nevertheless, the introduced artifacts could promote exploration in the way that the agent takes more random actions because of the ambiguity and randomness in the state representation. This can be a good trait when solving hard-exploration games like Montezuma's Revenge and Hero (Kapturowski et al., 2022) where the CWRCA agent obtained a higher mean evaluation score as depicted in Appendix D.

Table 1: **Mean standard deviation of actor logits**.

| Types of agents | CWCA | NA | SWA | CWRA | CWRCA |
|---|---|---|---|---|---|
| $\bar{\sigma}$ of actor logits | 4.0 | 2.91 | 4.29 | **2.65** | **2.65** |

**Exploration**    Following the same setup we have for the state representation study, we evaluate the degree of exploration based on the distributions of the logits of the actor network (a.k.a. the actor logits). Particularly in the context of PPO, the actions are sampled from a multinomial distribution, and the determinism of the action selection process depends on the distribution of the actor logits. For example, when logits are more evenly distributed (with lower variance), the action selection process becomes more random. Conversely, as the variance among logits increases, resulting in a more peaked distribution, the action selection process becomes more deterministic. In this study, we provide a simple and effective metric, i.e., the mean standard deviation of the actor logits to evaluate the degree of randomness of the policy. Specifically, for each agent, we compute the standard deviation of the actor logits per observation and then calculate the mean standard deviation over 10 sets of randomly selected observations. As depicted in Table 1, the CWRA and the CWRCA agents have the lowest mean standard deviation scores implying that these two agents are more exploratory than other agents at the three million time step. This could explain why both agents chose the less optimal action 'NOOP' as illustrated in Figure 6. Based on our observations from the interpretability study, it is evident that the inductive biases of the self-attention modules can influence the agent's sample efficiency in terms of state representation and exploration.

## 6 CONCLUSION

In this research, we investigated the fundamental operation of the self-attention mechanism in visual-based DRL settings. Specifically, we designed various self-attention modules by permuting the dimensions where the scaled dot-product operation is applied. We integrated the proposed designs with the PPO algorithm and evaluated their sample efficiency using the ALE benchmark. Our results indicate that different self-attention modules affect the agent's learning process differently, primarily due to the unique inductive bias of each self-attention module and the game mechanics. To understand how self-attention modules influence the sample efficiency of an agent, we perform an interpretability study through the lens of state representation and exploration. Our initial observations revealed that self-attention modules can generate artifacts that subtly influence the interplay between feature extraction, action selection, and reward collection. We believe that this work has made certain contributions to the ongoing efforts in optimizing DRL architectures, offering insights into the mechanisms that can enhance their performance in the low-data regime.

In the future, self-attention modules proposed in this work could be integrated and evaluated with other DRL algorithms and frameworks such as value-based RL algorithms and model-based RL respectively. It could also be interesting to combine various self-attention modules adaptively, especially in the context where the environment dynamics are unknown. Another promising research direction would be designing new self-attention or hybrid-attention mechanisms to enable more efficient and effective learning agents.

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

## A  NETWORK ARCHITECTURE

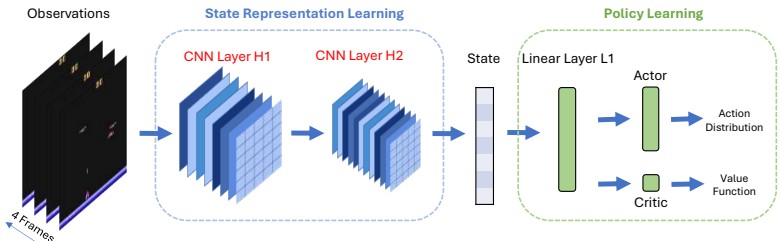

Figure 7: **Network architecture of PPO** (Schulman et al., 2017). The input observations comprise four consecutive frames of the game-play with each frame having a size of $84 \times 84$. The frames are processed by the first CNN layer H1 which contains 16 kernels with each kernel having a size of $8 \times 8$ and a stride of 4. The feature maps generated by H1 are subsequently processed by the second CNN layer H2 which has 32 $4 \times 4$ kernels with a common stride of 2. The feature maps generated by H2 are then flattened before being passed to the linear layer L1 of size 256. The outputs of L1 are forwarded to the actor network and the critic network for action selection and value estimation respectively. In this work, all the proposed self-attention modules are inserted between H1 and H2 to investigate their impacts on sample efficiency against the PPO baseline. The ReLU activation layer (Agarap, 2018) after each CNN and linear layer is not drawn explicitly in this figure.

## B  HYPERPARAMETERS

Table 2: **PPO hyperparameters**. $\alpha$ is linearly annealed from 1 to 0 over the entire training period. We used the default values from the Stable-Baselines3 v2.0.0 (Raffin et al., 2021) for hyperparameters not listed here.

| Parameter | Value |
|---|---|
| No. of parallel environments (n_envs) | 16 |
| Horizon (n_steps) | 128 |
| No. of epochs (n_epochs) | 3 |
| Minibatch size | $16 \times 16$ |
| Total timesteps (n_timesteps) | $1e7$ |
| Frame skipping | 4 |
| Frame stacking | 4 |
| Max no. of no-ops | 30 |
| Action repeat probability | 0 |
| Learning rate | $2.5 \times 10^{-4} \times \alpha$ |
| Clipping parameter | $0.1 \times \alpha$ |
| Value function coefficient | 1 |
| Entropy coefficient | 0.01 |
| Seeds | 0, 1, 10, 42, 1234 |

# C  PROBABILITY OF IMPROVEMENT

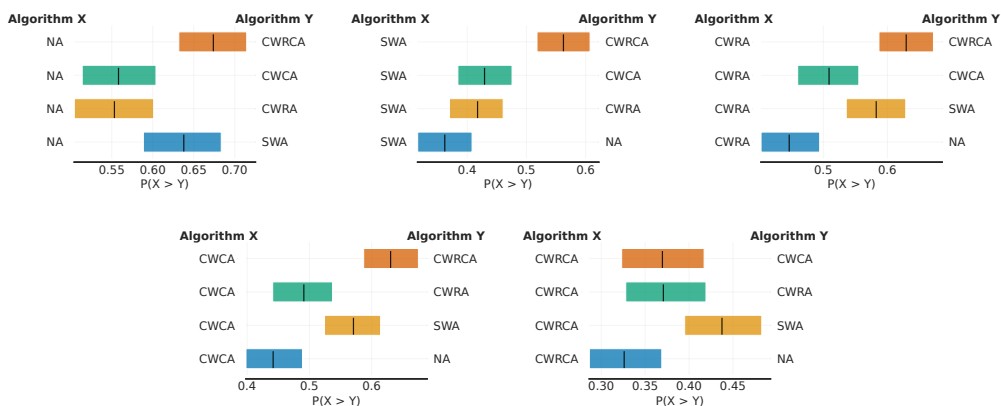

Figure 8: **Probability of improvement** (Agarwal et al., 2021). From top to bottom and from left to right, we demonstrate algorithm X's average probability of improvement over the other algorithms where algorithm X is represented by the NA, SWA, CWRA, CWCA, and the CWRCA agent respectively. Based on the top left sub-figure, the PPO baseline is more likely to outperform any self-attention models in a randomly selected game based on the mean evaluation scores. However, the chance of outperforming the CWRA and the CWCA agents by the PPO agent in any game is much less certain, implying that the self-attention models could perform better in certain games. In addition, IQM shown in Figure 2 serves as a more robust aggregate metric for sample efficiency.

## D  PERFORMANCE PER GAME

Table 3: **Performance per game**. In total, 56 games are evaluated over 10 million time steps across 5 seeds with all games having the 'NoFrameskip-v4' suffix in their environment IDs. The sample mean and standard error are computed using the mean evaluation score over the entire evaluation period across 5 runs. The 'winner' of each game is highlighted in bold based on the highest sample mean. Although the baseline agent has the highest number of wins, the combined impact of self-attention models (33 wins) is nontrivial and it is worth investigating how the inductive bias of each self-attention module influences the performance of the agent in different environments.

| Game | NA | SWA | CWRA | CWCA | CWRCA |
|---|---|---|---|---|---|
| Alien | 713.25 ± 23.93 | 783.05 ± 33.57 | **820.38 ± 55.90** | 796.17 ± 25.60 | 761.62 ± 27.09 |
| Amidar | 233.81 ± 17.10 | 194.72 ± 7.87 | 249.74 ± 18.04 | 219.80 ± 19.40 | **258.99 ± 24.00** |
| Assault | 1206.73 ± 81.79 | **1447.24 ± 145.57** | 1391.12 ± 150.19 | 1105.14 ± 30.39 | 1187.06 ± 141.38 |
| Asterix | **2190.20 ± 62.24** | 2115.20 ± 120.97 | 1850.88 ± 53.40 | 1881.16 ± 107.32 | 1751.08 ± 44.01 |
| Asteroids | **1694.04 ± 44.22** | 1613.50 ± 41.29 | 1519.31 ± 60.69 | 1540.75 ± 53.98 | 1515.91 ± 25.78 |
| Atlantis | **748152.32 ± 7500.59** | 706351.92 ± 8465.67 | 742176.56 ± 7568.71 | 717543.44 ± 15934.30 | 687103.60 ± 12713.27 |
| BankHeist | 282.67 ± 99.69 | 271.21 ± 96.26 | 291.63 ± 71.99 | **300.24 ± 79.89** | 289.53 ± 111.95 |
| BattleZone | **18500.80 ± 1010.68** | 15742.40 ± 831.53 | 17180.80 ± 1230.39 | 17376.00 ± 1091.82 | 14099.20 ± 1112.86 |
| BeamRider | **2473.49 ± 169.54** | 2035.45 ± 88.45 | 2394.04 ± 98.61 | 2238.17 ± 70.47 | 2001.04 ± 212.50 |
| Berzerk | 744.35 ± 28.95 | 783.36 ± 29.50 | **859.44 ± 11.41** | 738.88 ± 28.79 | 821.95 ± 30.29 |
| Bowling | 37.08 ± 1.39 | 42.24 ± 3.00 | 39.27 ± 3.15 | **43.77 ± 3.17** | 35.87 ± 2.61 |
| Boxing | 32.44 ± 1.67 | 27.90 ± 4.80 | 41.50 ± 2.62 | **44.44 ± 6.78** | 25.96 ± 3.52 |
| Breakout | **49.17 ± 2.54** | 38.55 ± 1.92 | 40.71 ± 2.99 | 42.61 ± 1.84 | 42.29 ± 5.29 |
| Centipede | 3171.62 ± 49.00 | 3103.86 ± 34.70 | 3107.74 ± 70.54 | 2980.11 ± 102.51 | **3231.28 ± 56.19** |
| ChopperCommand | 1795.84 ± 86.32 | 1614.56 ± 22.34 | **1909.92 ± 72.07** | 1609.76 ± 47.24 | 1536.64 ± 90.13 |
| CrazyClimber | 83603.20 ± 2103.07 | 79674.88 ± 2573.49 | 82909.68 ± 1900.91 | **83995.60 ± 2277.56** | 78999.12 ± 670.93 |
| Defender | 13075.88 ± 421.91 | 12774.72 ± 483.22 | 12650.36 ± 701.04 | **15035.48 ± 679.71** | 14171.84 ± 656.77 |
| DemonAttack | 4276.36 ± 148.32 | **4526.02 ± 245.87** | 4430.51 ± 376.03 | 4059.09 ± 63.70 | 3695.32 ± 103.82 |
| DoubleDunk | **-6.13 ± 0.31** | -6.78 ± 0.32 | -6.15 ± 0.22 | -6.20 ± 0.29 | -6.32 ± 0.14 |
| Enduro | **176.80 ± 43.35** | 162.52 ± 25.48 | 129.54 ± 32.08 | 112.26 ± 17.96 | 106.03 ± 35.66 |
| FishingDerby | -70.46 ± 3.11 | -78.18 ± 1.37 | -72.74 ± 2.99 | **-66.00 ± 2.19** | -71.96 ± 2.20 |
| Freeway | **29.23 ± 0.28** | 28.73 ± 0.43 | 23.74 ± 4.05 | 24.33 ± 5.44 | 23.24 ± 5.20 |
| Frostbite | 270.59 ± 2.60 | 268.18 ± 2.46 | 279.81 ± 3.50 | **676.94 ± 364.66** | 266.51 ± 3.07 |
| Gopher | 893.97 ± 21.91 | 896.82 ± 28.77 | **954.93 ± 21.44** | 913.07 ± 18.05 | 917.46 ± 9.25 |
| Gravitar | **328.68 ± 20.12** | 318.76 ± 18.17 | 295.28 ± 8.63 | 299.40 ± 9.79 | 261.36 ± 8.32 |
| Hero | 9045.84 ± 116.65 | 8435.23 ± 393.62 | 9153.70 ± 280.52 | 9071.00 ± 282.13 | **9877.38 ± 145.04** |
| IceHockey | **-4.78 ± 0.13** | -5.06 ± 0.19 | -4.93 ± 0.08 | -4.97 ± 0.14 | -4.90 ± 0.08 |
| Jamesbond | 609.08 ± 88.48 | 480.32 ± 14.36 | **693.60 ± 119.26** | 457.88 ± 14.61 | 452.44 ± 30.54 |
| Kangaroo | 1504.24 ± 272.96 | 1503.60 ± 181.72 | **1886.56 ± 291.60** | 1250.56 ± 103.63 | 1252.64 ± 292.68 |
| Krull | 5537.86 ± 196.97 | 4970.93 ± 149.33 | 5189.49 ± 107.28 | **5763.52 ± 166.26** | 5095.27 ± 185.98 |
| KungFuMaster | **17357.68 ± 700.29** | 17260.96 ± 1426.21 | 17050.72 ± 1425.88 | 17110.80 ± 725.67 | 13422.16 ± 1048.28 |
| MontezumaRevenge | 0.72 ± 0.49 | 0.40 ± 0.28 | 0.48 ± 0.18 | 0.48 ± 0.35 | **2.16 ± 1.25** |
| MsPacman | **772.44 ± 15.97** | 699.65 ± 10.00 | 717.33 ± 38.01 | 686.30 ± 23.43 | 669.47 ± 8.52 |
| NameThisGame | **5176.36 ± 79.77** | 4668.89 ± 81.98 | 5116.64 ± 81.12 | 4812.22 ± 223.28 | 4493.71 ± 178.70 |
| Phoenix | 4200.87 ± 103.45 | 4206.65 ± 185.15 | 4194.28 ± 52.70 | **4367.82 ± 92.50** | 4106.22 ± 142.62 |
| Pitfall | **-7.66 ± 1.37** | -16.36 ± 5.65 | -28.05 ± 11.92 | -10.73 ± 3.88 | -11.98 ± 1.80 |
| Pong | 9.91 ± 0.53 | 8.32 ± 0.74 | 7.30 ± 1.60 | **12.64 ± 0.43** | -0.02 ± 4.01 |
| PrivateEye | 93.06 ± 1.64 | 87.12 ± 9.55 | 88.90 ± 2.55 | 84.64 ± 2.99 | **109.90 ± 22.71** |
| Qbert | **1594.34 ± 74.58** | 1228.14 ± 63.00 | 1467.60 ± 87.12 | 1425.32 ± 128.60 | 1128.26 ± 93.87 |
| Riverraid | 4098.34 ± 319.24 | 4464.29 ± 101.54 | **4548.46 ± 177.25** | 4468.96 ± 268.44 | 3822.38 ± 252.04 |
| RoadRunner | **17679.60 ± 1207.69** | 14792.88 ± 1527.42 | 15625.60 ± 1066.88 | 15596.96 ± 541.54 | 13924.72 ± 1252.82 |
| Robotank | **15.76 ± 0.87** | 14.44 ± 0.64 | 14.27 ± 0.63 | 13.16 ± 0.64 | 10.28 ± 0.92 |
| Seaquest | **865.14 ± 2.13** | 845.89 ± 3.02 | 854.10 ± 3.70 | 851.44 ± 1.49 | 843.82 ± 4.34 |
| Skiing | -28852.89 ± 549.36 | -21709.29 ± 4541.60 | -21695.07 ± 4566.52 | -17406.93 ± 4598.34 | **-13266.78 ± 3785.99** |
| Solaris | **2344.58 ± 47.87** | 2332.13 ± 70.77 | 2199.54 ± 43.66 | 2278.88 ± 43.55 | 2337.78 ± 78.62 |
| SpaceInvaders | 515.05 ± 8.99 | **532.38 ± 13.09** | 504.85 ± 9.32 | 516.86 ± 16.91 | 487.15 ± 5.81 |
| StarGunner | 8952.08 ± 569.54 | 8824.72 ± 509.83 | 9063.12 ± 587.46 | **9602.56 ± 468.26** | 8372.16 ± 919.90 |
| Tennis | -16.09 ± 2.41 | **-11.09 ± 1.75** | -16.20 ± 1.74 | -13.36 ± 2.31 | -11.90 ± 0.85 |
| TimePilot | **4938.48 ± 148.33** | 4501.84 ± 154.27 | 4330.56 ± 194.76 | 4789.36 ± 145.20 | 4369.52 ± 190.30 |
| Tutankham | **160.79 ± 1.80** | 160.08 ± 2.57 | 156.58 ± 2.36 | 158.20 ± 1.88 | 156.87 ± 3.24 |
| UpNDown | 49361.81 ± 15012.83 | 35094.31 ± 1432.76 | 59758.81 ± 17960.40 | **64822.30 ± 15476.49** | 23096.82 ± 2146.71 |
| Venture | **13.12 ± 4.82** | 5.28 ± 2.34 | 8.16 ± 6.26 | 4.16 ± 2.06 | 3.68 ± 2.26 |
| VideoPinball | 25318.44 ± 287.01 | **25979.93 ± 654.44** | 25669.86 ± 885.77 | 24888.83 ± 899.18 | 25354.74 ± 833.24 |
| WizardOfWor | 3415.92 ± 168.71 | 3100.56 ± 229.14 | **3819.76 ± 145.49** | 3475.28 ± 294.38 | 3504.88 ± 134.35 |
| YarsRevenge | 13977.03 ± 1935.09 | 13141.56 ± 357.95 | 10376.39 ± 1936.01 | **15025.78 ± 523.89** | 13697.67 ± 582.27 |
| Zaxxon | 5381.20 ± 603.69 | 4293.04 ± 1237.78 | 5872.40 ± 619.70 | **6504.00 ± 498.32** | 5719.60 ± 828.21 |
| **No. of wins** | 23 | 5 | 8 | 14 | 6 |

# E  GAMES WON BY EACH AGENT

The list of games won by each agent is curated based on the mean performance as shown in Appendix D. Although the sample mean is calculated from a few runs which presents a certain degree of uncertainty as indicated by the standard error, we believe that there could exist a subtle correlation between the inductive bias of each self-attention module and the game mechanics. In other words, we aim to discover the commonality among all the games won by a particular agent which could help us understand why such an agent can learn faster in these games but not in others.

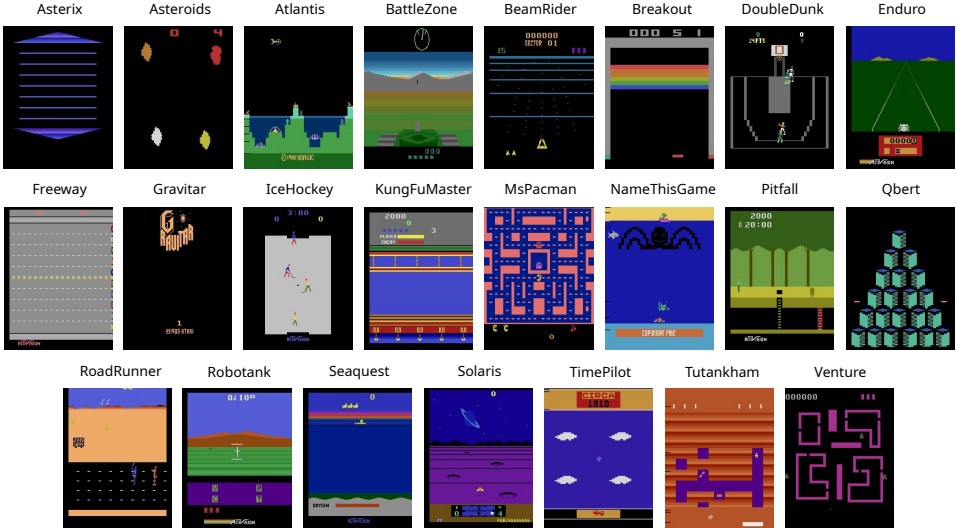

Figure 9: **Games won by the NA agent**. The NA agent depicted in Appendix A is the PPO baseline without the self-attention module. The inductive bias of the state representation block primarily arises from the CNN layers. Overall, there appears to be limited commonality among the games won by the NA agent, likely due to its broad feature extraction capabilities from CNN.

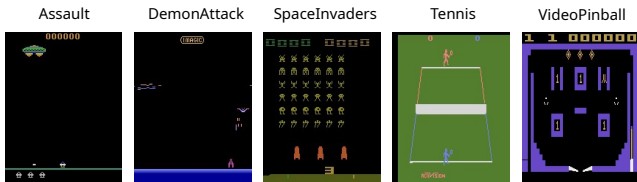

Figure 10: **Games won by the SWA agent**. Compared with the baseline, the SWA agent contains an additional self-attention module that performs the dot product operation over the row and column dimensions and repeats it along the channel dimension. Although only 5 games are won by the SWA agent, it seems that games with more static backgrounds (e.g., no scrolling of the game scene) and fewer distinctive objects can be 'taken advantage of' by the SWA agent. For instance, both VideoPinball and Tennis feature simpler backgrounds with fewer moving elements.

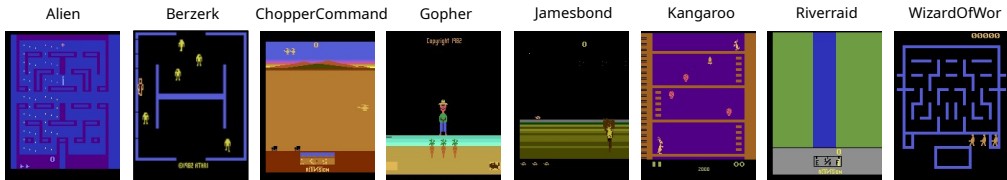

Figure 11: **Games won by the CWRA agent**. The self-attention module possessed by the CWRA agent carries out the dot product operation over the channel and row dimensions and repeats it along the column dimension. We hypothesize that dynamics along the column (width) dimension could be 'captured and utilized' by the CWRA module naturally. For example, games with rewarding objects moving horizontally such as Gopher, and horizontally scrolling games like ChopperCommand and Jamesbond are won by the CWRA agent.

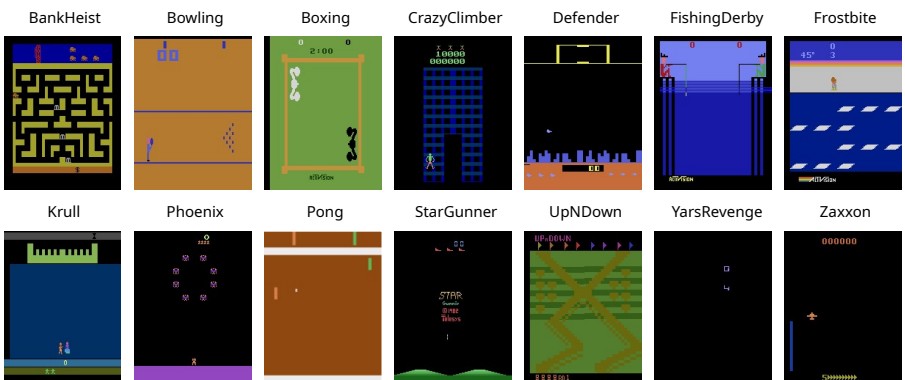

Figure 12: **Games won by the CWCA agent**. In contrast to the CWRA agent, the CWCA module implements the dot product operation over the channel and column dimension and repeats it along the row dimension. Intuitively, we assume that dynamics along the row (height) dimension could be 'leveraged' by the CWCA module more effectively. Following this assumption, we observe that games with rewarding objects moving vertically such as FishingBerby, Krull, and Pong as well as vertically scrolling games like CrazyClimber, UpNDown, and Zaxxon are won by the CWCA agent.

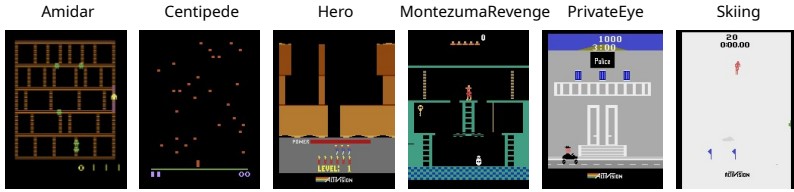

Figure 13: **Games won by the CWRCA agent**. Intending to combine the advantages of the CWRA and the CWCA modules, the CWRCA agent integrates both modules via an element-wise summation operation. This could enable it to attend to dynamics along all dimensions. On the one hand, attending to all dimensions could over-complicate the state representation and the agent may spend more effort disentangling the patterns which slows down the learning process, like in the case of the Pong game. On the other hand, attending to all dimensions could encourage exploration due to the high entropy (e.g., noise) injected into the state space. This could increase the agent's learning efficiency, especially in hard-exploration games like Montezuma's Revenge and Hero.

# F  LEARNING CURVES PER GAME

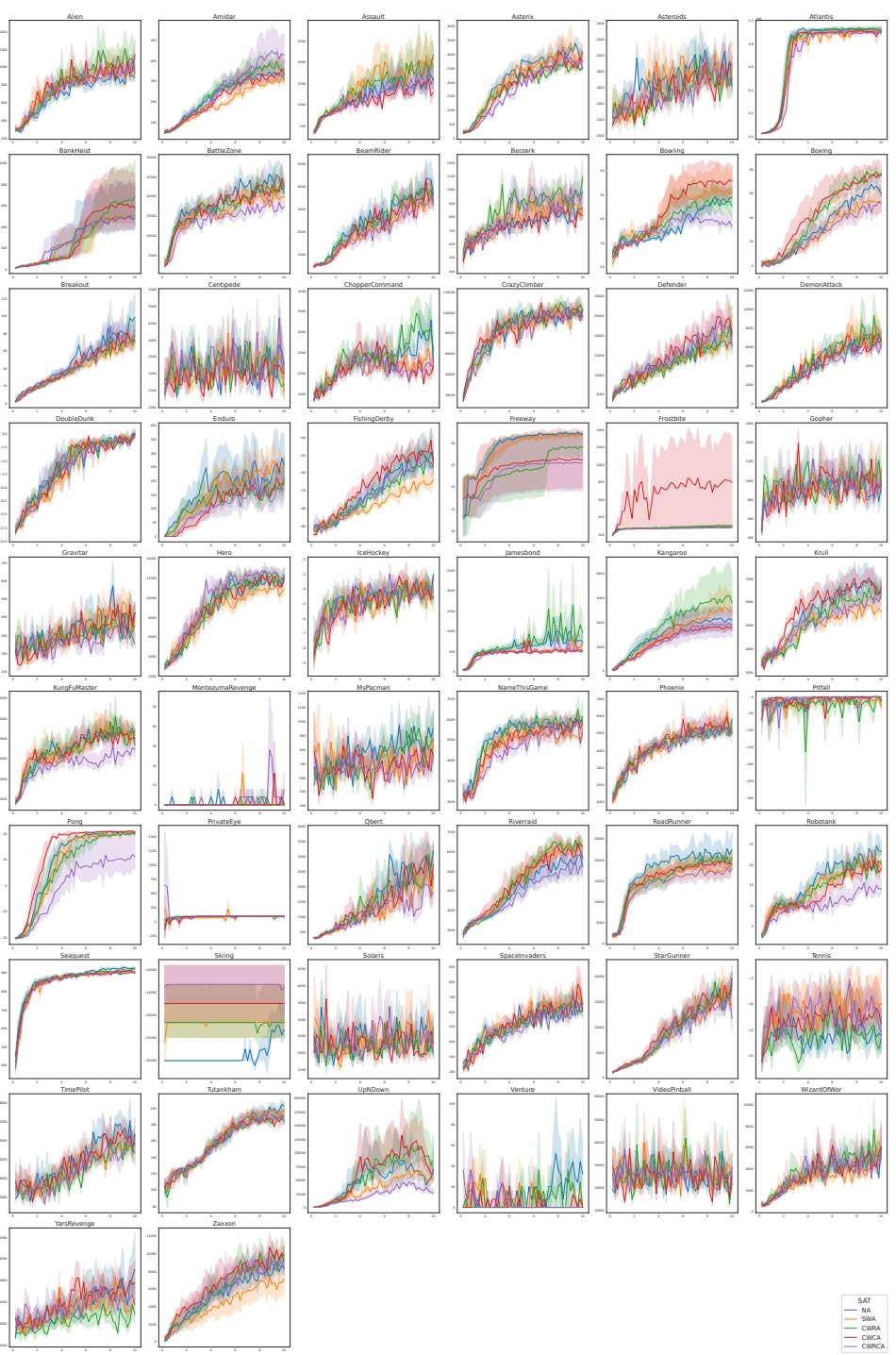

Figure 14: **Learning curves per game**. The solid line indicates the mean performance, while the shaded region represents the 95% confidence interval over 5 runs. The term 'SAT' in the legend field stands for Self-Attention Type which is detailed in Section 4.

# G STATE REPRESENTATION AND EXPLORATION

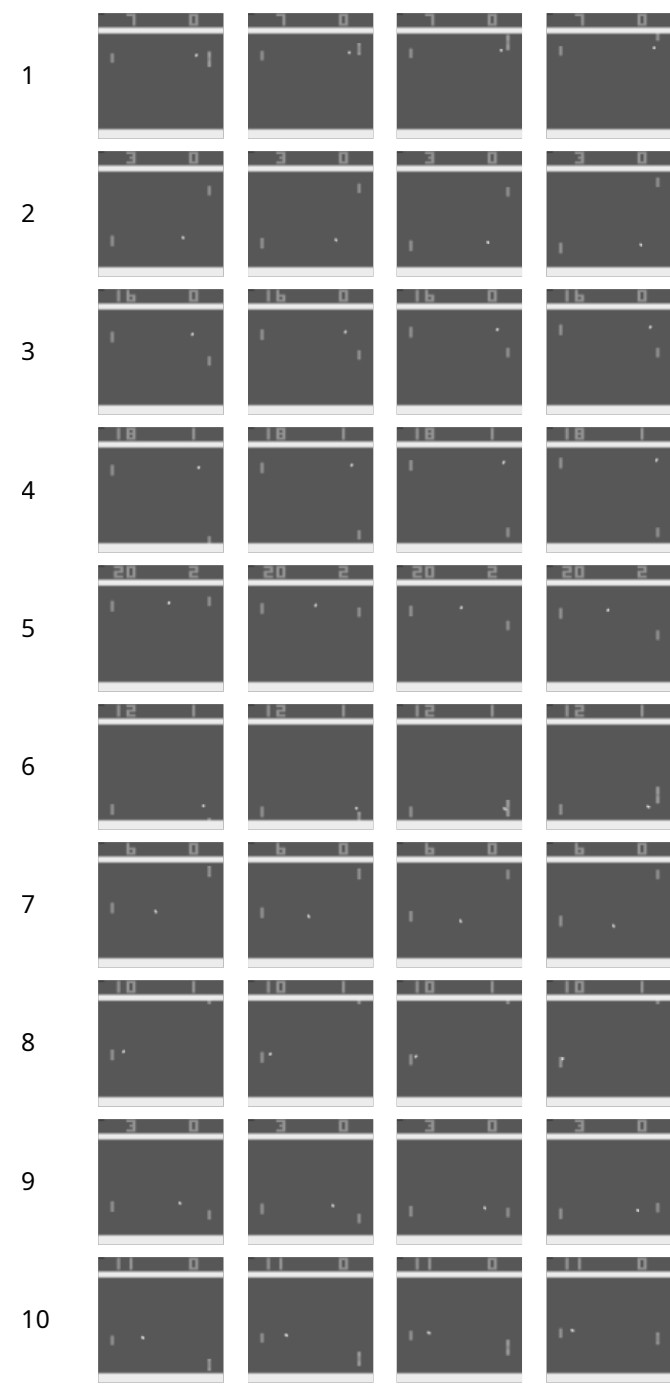

Figure 15: **10 sets of random observations based on the Pong game**. Each set of observations comprises 4 consecutive frames and all sets of observations are generated by a random policy. The seventh set of observations is used in the interpretability study.

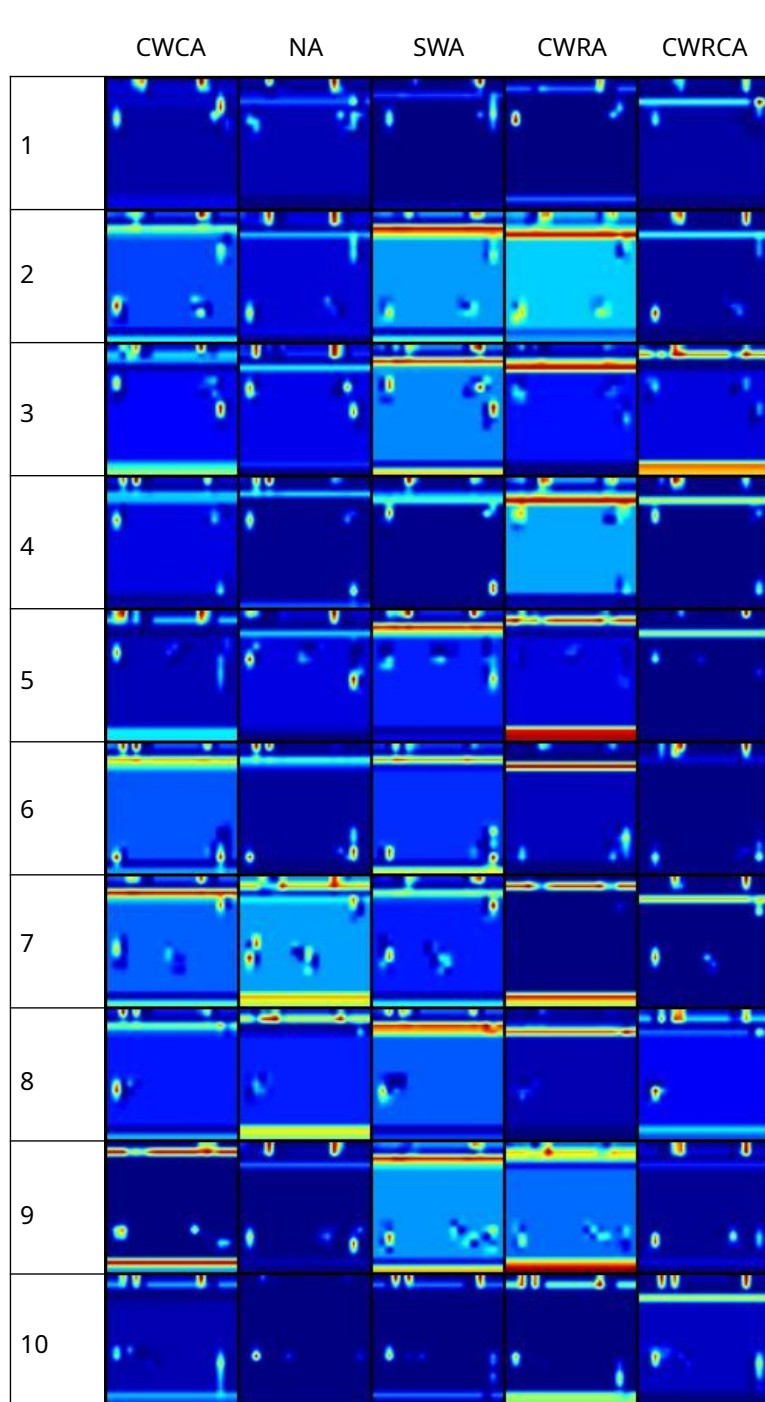

Figure 16: **10 sets of heatmaps at the first CNN layer based on the Pong game**. Each row pictures the heatmaps of all agents at the first CNN layer based on the observations in Figure 15 and the best actions in Table 4. The computation of the heatmap is illustrated in Section 5.3. In general, all agents can correlate the key objects in the scene with their actions. The SWA and the CWRA agents tend to highlight the walls more often than other agents.

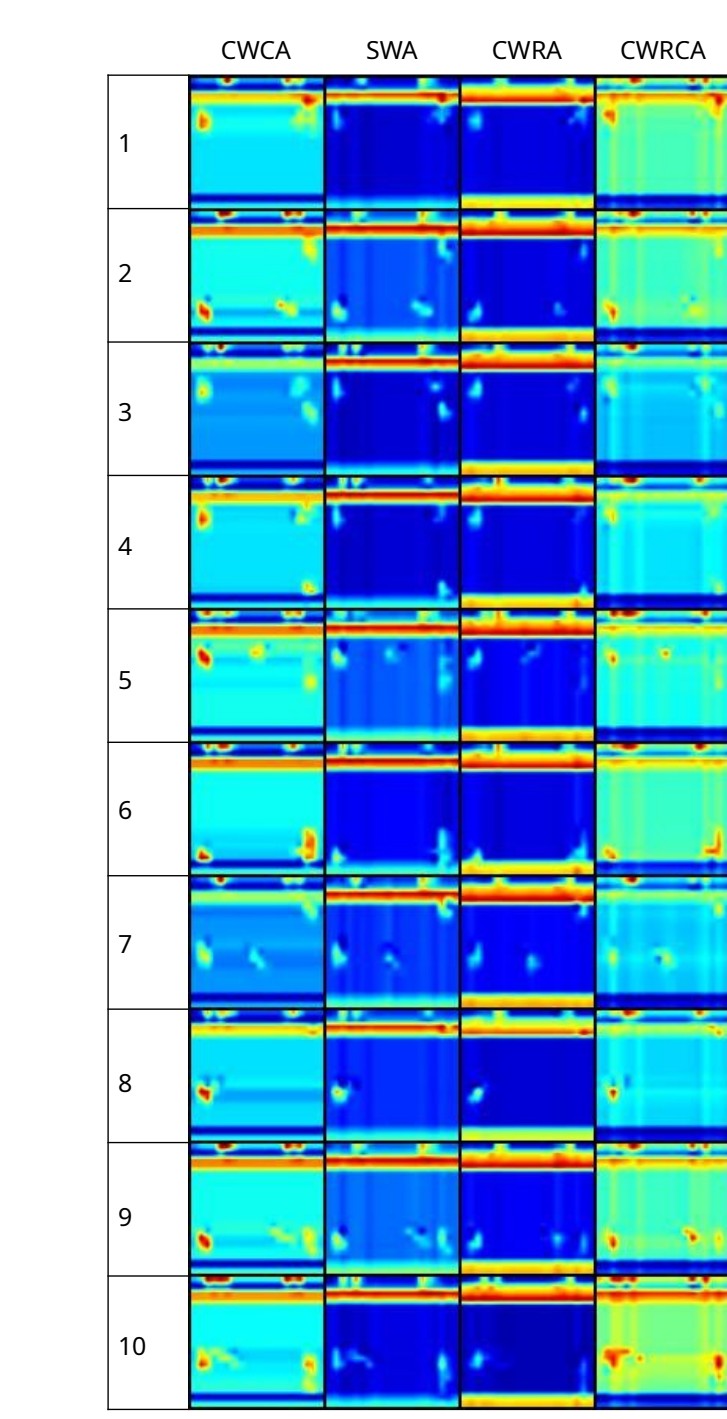

Figure 17: **10 sets of attended feature maps based on the Pong game**. Each row depicts the attended feature maps of all self-attention-enabled agents based on the observations in Figure 15. It can be seen that different self-attention modules can create different artifacts. The artifacts created by the SWA module resemble vertical bars whereas artifacts generated by the CWCA agent resemble horizontal bars. Intuitively, the CWRCA module creates both horizontal and vertical bar-like artifacts.

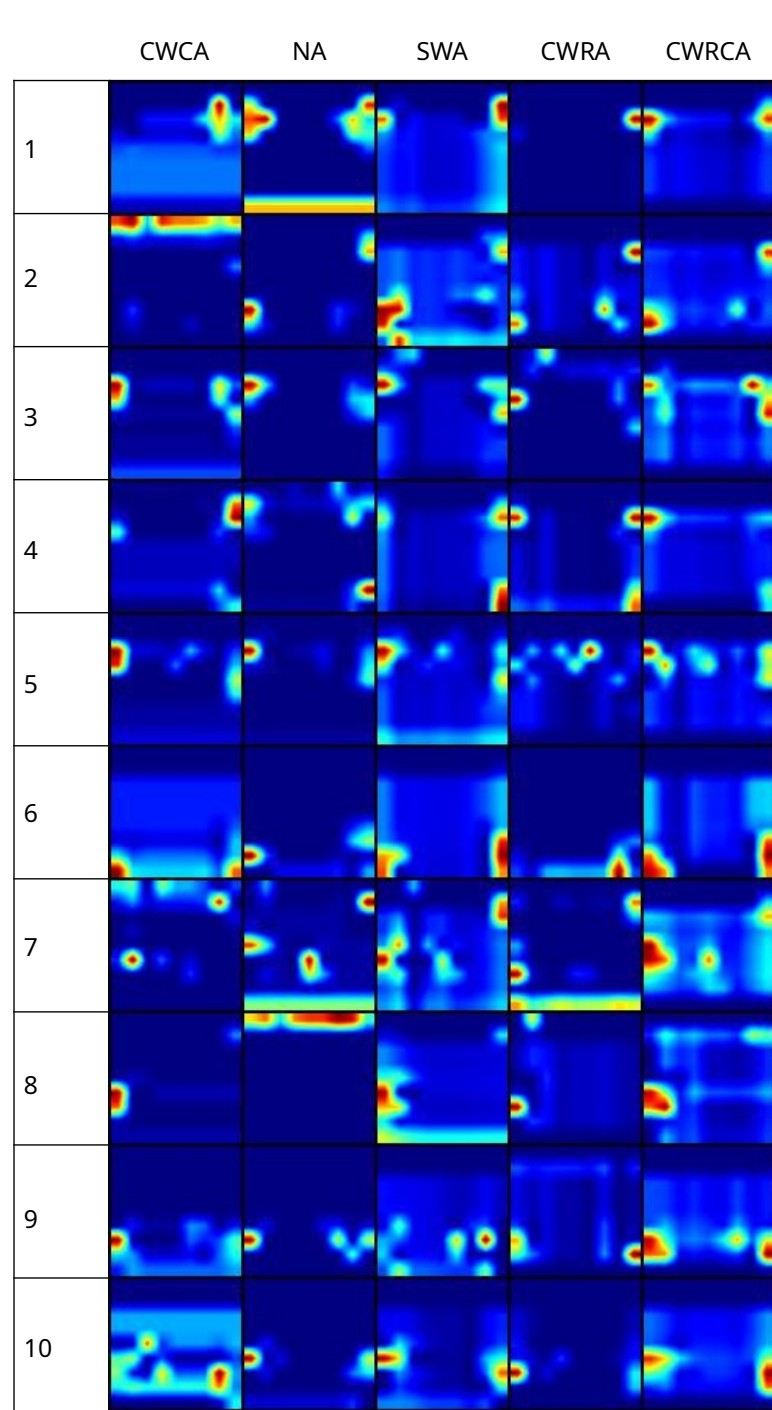

Figure 18: **10 sets of heatmaps at the second CNN layer based on the Pong game**. A key distinction between the heatmaps from the first and second CNN layers is the emergence of artifacts generated by the self-attention modules since the attended feature maps serve as inputs to the second CNN layer. The presence of the artifacts could play a subtle role in influencing the agent's learning efficiency in terms of state representation and exploration which is discussed in Section 5.3.

Table 4: **10 sets of best actions based on the Pong game**. Each row represents the agent's best action (deterministic = True) corresponding to the input observations as shown in Figure 15. Model checkpoints are selected at the 3 million time step as detailed in Section 5.3.

|  | CWCA | NA | SWA | CWRA | CWRCA |
|---|---|---|---|---|---|
| 1 | NOOP | LEFT | FIRE | NOOP | NOOP |
| 2 | LEFT | LEFTFIRE | LEFTFIRE | LEFT | LEFTFIRE |
| 3 | LEFT | LEFTFIRE | LEFTFIRE | LEFT | LEFTFIRE |
| 4 | NOOP | LEFTFIRE | FIRE | RIGHT | NOOP |
| 5 | LEFT | LEFTFIRE | LEFTFIRE | LEFT | LEFTFIRE |
| 6 | RIGHT | RIGHT | RIGHTFIRE | RIGHT | RIGHT |
| 7 | LEFTFIRE | LEFTFIRE | LEFTFIRE | NOOP | NOOP |
| 8 | LEFT | LEFT | LEFTFIRE | LEFT | LEFTFIRE |
| 9 | LEFTFIRE | LEFTFIRE | LEFTFIRE | LEFT | LEFTFIRE |
| 10 | FIRE | RIGHT | RIGHT | NOOP | NOOP |

