# OpenReview forum: "Investigating Self-Attention: Its Impact on Sample Efficiency in Deep Reinforcement Learning"
_ICLR.cc/2025/Conference — ICLR 2025 Conference Withdrawn Submission_

### Official Review · Reviewer_dbpQ · 2024-10-22

**Soundness:** 3
**Presentation:** 3
**Contribution:** 2
**Rating:** 5
**Confidence:** 4

**Summary:**

This paper investigates the impact of self-attention when improving the sample efficiency in DRL. Specifically, the investigation focuses on how different types of scaled dot-product attention affect the performance in ALE. The methods are categorized by over which channels the dot product is applied, including SWA, CWRA, CWCA and CWRCA. In particular, the author discovered the effect of inductive biases of the self-attention modules, such as attending to objects movement horizontally or vertically can be rewarded in different game environments. The author conclude that self-attention modules have different effect to the interplay between the inductive bias and the game mechanics.

**Strengths:**

1. The topic is quite interesting, as it connects NLP and DRL, and self-attention has also attracted lots of attention in developing more advanced RL algorithms nowadays.

2. The setup of experiments is quite thorough and well-thought, especially with the evaluation metrics and state presentations.

3. The results are quite interesting and give us new perspective about the importance of self-attention. Meanwhile, I found te explanations of those results satisfactory.

**Weaknesses:**

1. The author did not give information about the choices of hyper-parameters or any ablation study regarding the network structure. In the experiment section, only the experimental setup and the evaluation methodology are discussed. This can be a problem.

2. Following point 1, what kinds of steups for self-attention modules have you tested? Have you tried to vary the number of self-attention layers and the heads? I think it is crucial to study all different aspects of MHSA as well. This is because MHSA is more applicable in NLP and Vision Transformers than simplex self-attention operations.

**Questions:**

1. Can you disclose the hyper-parameter choices?

2. Can you give more explanations on what the current results imply towards MHSA? Explain the rationale for focusing on single-head attention and whether the findings would generalize to MHSA.

---

> ### Author Response · Authors · 2024-12-02
>
> Dear Reviewer dbpQ,
>
> Thank you for your time, valuable feedback, and insightful comments! We have revised our paper and would like to address the weaknesses and the questions as follows:
>
> **Responses to the weaknesses**
>
> >**W1**: The author did not give information about the choices of hyper-parameters or any ablation study regarding the network structure.
>
> We thank the reviewer for the constructive feedback. We have included the hyperparameters in **Appendix B** in the revised version of the paper. In addition, we have updated the **supplementary material** to open-source all the code used in our work for reproducibility and future enhancement. In terms of the ablation study on the network architecture, we believe that the experiment design is self-explanatory in the sense that the only change between each model architecture is the dimensions where the scaled dot-product attention is applied while all the hyperparameters are fixed throughout the experiment. Therefore, when comparing each self-attention design with the baseline model, there is only one controlled variable to consider, i.e., the dimensions where the dot product is applied. Hence, we believe that the ablation study is somewhat inherent in our work when we compare each self-attention model with the baseline model.
>
> **W2**: What kinds of setups for self-attention modules have you tested? Have you tried to vary the number of self-attention layers and the heads?
>
> We thank the reviewer for posing these questions and considering the implications of MHSA. As for the setup, at a very high level, all self-attention modules shown in **Figure 1** are integrated into the state representation learning block of the PPO algorithm whose full model architecture is shown in **Appendix A** in the revised version of the paper. Specifically, each self-attention module is sandwiched between the first and the second CNN layers. In contrast to the setup in NLP and ViT, the feature maps at the 1st CNN layer are treated as input tokens in our context. In other words, the input tokens are 3D tensors instead of 2D matrices. There is a total of 16 20x20 feature maps at the 1st CNN layer where we can imagine each feature map captures some part of the state and all the 16 feature maps represent a single (full) state (in RL, the state is the only input to the model and is mapped to an action by the policy). The query, key, and value are generated using 3 1x1 CNN layers respectively with each CNN layer containing 16 16x1x1 CNN kernels. Each CNN kernel transforms the entire set of feature maps at the 1st CNN layer (the state) into a new feature map (a new representation of the state), and all 16 CNN kernels generate 16 different representations of the states. This process is repeated during the query, key, and value formations and all of them have a shape of 16x20x20. These 3D $Q$, $K$, and $V$ are then used to perform the scaled dot-product attention over different dimensions. In this vein, we would prefer to think of our setup as closer to the MHSA since the number of heads in our context can be regarded as the number of 16x1x1 CNN kernels and can also be regarded as the number of different state representations in the final $Q$, $K$, and $V$. Interestingly, we also see the same analogy being made between the MHSA and multiple kernels in CNN in [1]. Although we are more clingy to thinking about our configuration as more similar to the MHSA, we did not explicitly indicate it in our paper because it does depend on your interpretation. In this regard, we are open to other interpretations and welcome any feedback.
>
>
> **Responses to the questions**
>
> >**Q1**: Can you disclose the hyper-parameter choices?
>
> Please refer to the response to the weakness **W1**.
>
> >**Q2**: Can you give more explanations on what the current results imply towards MHSA? Explain the rationale for focusing on single-head attention and whether the findings would generalize to MHSA.
>
> Please refer to the response to the weakness **W2**.
>
> [1] https://sebastianraschka.com/pdf/lecture-notes/stat453ss21/L19_seq2seq_rnn-transformers__slides.pdf (slide 47 on Multi-Head Attention)

---

> > ### Comment · Reviewer_dbpQ · 2024-12-02
> > **Thank you**
> >
> > I thank the author for answering my questions. The score remains unchanged.

---

### Official Review · Reviewer_h3CB · 2024-11-02

**Soundness:** 2
**Presentation:** 2
**Contribution:** 1
**Rating:** 3
**Confidence:** 4

**Summary:**

This work investigates the performance differences on ALE games with different self-attention operations for visual reinforcement learning. The self-attention operations include (1) direct self-attention on the feature map, (2) self-attention on the height dimension, (3) self-attention on the width dimension, and (4) the sum of (2) and (3). The conclusion is that different attention operations have different impacts on different games, which is brought about by different game mechanics.

**Strengths:**

1. This paper investigates an interesting topic after Manchin et al., examining whether different attention operations can impact visual DRL learning differently.

2. This paper evaluates 56 games within the ALE benchmark, ensuring the broad applicability of the results, and uses stratified bootstrap confidence intervals for reliable assessment. Compared with Manchin et al., the experiment uses 56 games and 5 seeds, which enhances the experimental results.

3. The paper designs and contrasts four types of self-attention modules (SWA, CWRA, CWCA, CWRCA), revealing how each module’s inductive biases affect learning efficiency in specific environments.

4. The paper is brief and straightforward.

**Weaknesses:**

1.  After reading the entire paper, I still struggle to understand the connection between the terms “sample efficiency” and “attention operation” in this paper. To my understanding, the attention operation only makes differences in network structures. It makes sense that different network structures bring different learning performance. However, “sample efficiency” relates to choosing different transition pairs from the replay buffers or the experience batches. Could you explain more about this part?

2.  The related work section should contain more references. Three papers are not enough; you should investigate and survey more.

3. The types of self-attention are far more varied than different channel operation sequences. More investigations are required if the author aims to make significant and fundamental contributions to the ongoing efforts to optimize DRL architecture.

4. This paper's insights are poorly developed, lack theoretical analysis, and rely merely on limited observations. It fails to provide adequate insights for the community. E.g., “Different attention operations have different impacts on different games” cannot guide the community. Conclusions like “height dimension self-attention provides more performance improvement on vertically moving games” would be more beneficial.

[1] Mott, Alexander, et al. "Towards interpretable reinforcement learning using attention augmented agents." In *Advances in Neural Information Processing Systems*, 2019.

**Questions:**

1. In Figure 6, methods lacking an attention mechanism seemingly learn critical environmental information effectively. Conversely, variants incorporating attention mechanisms appear to have limited learning capabilities, often failing to capture essential details or maintain precise focus. Could you provide explanations for this phenomenon?
2. Figure 7 shows that the mean and standard deviation of the logits for both CWRA and CWRCA are the lowest, which indicates a less exploratory behavior due to a more uneven logit distribution. Are the conclusions presented in the paper incorrect?
3. Could you analyze the causes and conditions under which artifacts occur? Are there potential ways to eliminate these artifacts? For example,  Related work like [2] explores artifacts in vision. Can you offer similar in-depth insights?
4. Beyond the scaled dot-product operation discussed in this paper, could other aspects of attention affect sample efficiency?

[2] Darcet, Timothée, et al. "Vision transformers need registers." In *International Conference on Learning Representations*, 2024.

---

> ### Author Response · Authors · 2024-12-01
> **Responses to the weaknesses**
>
> Dear Reviewer h3CB,
>
> Thank you for your time, valuable feedback, and insightful comments! We have revised our paper and would like to address the weaknesses as follows:
>
> **Responses to the weaknesses**
>
> >**W1**: However, “sample efficiency” relates to choosing different transition pairs from the replay buffers or the experience batches. Could you explain more about this part?
>
> We thank the reviewer for posing this question. Sample efficiency in RL generally refers to the number of interactions with the environment that an agent needs to achieve a specific level of performance [1]. It can be measured in two ways. When the target performance is given, the agent that requires the fewest interactions with the environment to reach the target performance has the highest sample efficiency. When the number of interactions is capped, the agent that achieves the highest performance has the highest sample efficiency. In most sample efficiency benchmarks, the number of interactions is capped at a certain threshold, and the mean performance of the agents is used to compare sample efficiency. In our work, the total number of interactions, i.e., the training duration is fixed at 10 million timesteps, and the mean evaluation score is used to assess the agent's sample efficiency. For RL algorithms that use the replay buffer, although the replay ratio can be adjusted to improve sample efficiency, the sample efficiency itself still refers to the actual number of interactions with the environment. We hope this explanation helps.
>
> >**W2**: The related work section should contain more references.
>
> We thank the reviewer's feedback and included more references in the revised version of the paper.
>
> >**W3**: More investigations are required if the author aims to make significant and fundamental contributions to the ongoing efforts to optimize DRL architecture.
>
> We acknowledge that the self-attention module designs proposed in our work have certain limitations, and we recognize that the claims regarding the paper's contributions can be refined to more accurately reflect the scope of the work.
>
> >**W4**: This paper's insights are poorly developed, lack theoretical analysis, and rely merely on limited observations. It fails to provide adequate insights for the community.
>
> We appreciate the reviewer's recommendation regarding the conclusion and acknowledge that the contributions of this paper can be further improved by providing more conclusive statements and offering more guidance to the community.
>
> **References**
>
> [1] Towards Sample Efficient Reinforcement Learning (Yu, 2018)

---

> ### Author Response · Authors · 2024-12-01
> **Responses to the questions**
>
> Dear Reviewer h3CB,
>
> Thank you for your time, valuable feedback, and insightful comments! We have revised our paper and would like to address the questions as follows:
>
> **Responses to the questions**
>
> >**Q1**: In Figure 6, methods lacking an attention mechanism seemingly learn critical environmental information effectively. Conversely, variants incorporating attention mechanisms appear to have limited learning capabilities, often failing to capture essential details or maintain precise focus. Could you provide explanations for this phenomenon?
>
> We appreciate the reviewer's insightful observations and have identified that the primary cause of this phenomenon is the Grad-CAM visualization technique. To be more specific, the heatmaps generated via Grad-CAM are biased or sensitive to the "best" action chosen by the agent, making them less suitable for visualizing the complete state information. A more effective approach for state visualization would be to use the rectified feature maps generated during the forward pass which are insensitive or unbiased to the actions. Although we did not manage to include the updated results in the latest version of the paper, we find this new approach significantly better in terms of the accuracy of the state representations. We hope this new approach will benefit the community as well.
>
> >**Q2**: Figure 7 shows that the mean and standard deviation of the logits for both CWRA and CWRCA are the lowest, which indicates a less exploratory behavior due to a more uneven logit distribution. Are the conclusions presented in the paper incorrect?
>
> We acknowledge the potential confusion caused by the term "mean standard deviation of the actor logits" which aims to measure the amount of variation of the logits about its mean. For example, when logits are more evenly distributed (with lower variance, think of a uniform distribution where logits are almost equal), the action selection process becomes more random (exploratory). Conversely, as the variance of logits increases, resulting in a more peaked distribution (like a Gaussian or skewed distribution), the action selection process becomes more deterministic (exploitative). Understand that the term standard deviation is often associated with the "spread" of the distribution, making it less intuitive for measuring randomness. We propose a new metric, namely, the action entropy to measure the randomness of the agent's actions. Although we did not include the updated results in the latest version of the paper, we find the new metric more intuitive, and mathematically sound and provides the same insight where the CWRCA agent is more exploratory. We hope this new metric will benefit the community as well.
>
> >**Q3**: Could you analyze the causes and conditions under which artifacts occur? Are there potential ways to eliminate these artifacts? For example, Related work like [2] explores artifacts in vision. Can you offer similar in-depth insights?
>
> 1) We appreciate the interesting questions posed by the reviewer. The main causes of the artifacts are twofold. Firstly, the CNN module and the self-attention module form a complementary state representation learning system where the self-attention module tries to highlight information missed or not captured by the CNN module. Secondly, each self-attention module has a unique inductive bias due to the dimensions where the dot product is applied. In general, self-attention modules that perform dot product operations involving the height dimension can generate vertical block attention (vertical bar-like artifact), while those involving the width dimension can generate horizontal block attention (horizontal bar-like artifact).
>
> 2) To remove artifacts, the simplest way is to remove the corresponding self-attention module from the model architecture. However, in our work, the artifacts produced by the self-attention modules behave as a double-edged sword. Instead of figuring out how to remove them, we explore how to benefit the most from the dimensional bias of the self-attention modules.
>
> >**Q4**: Beyond the scaled dot-product operation discussed in this paper, could other aspects of attention affect sample efficiency?
>
> We thank the reviewer for posing this question. In this work, we focus on the dimensional bias of self-attention in a particular setup where the input tokens have more than 2 dimensions. In a broader context, the types of attention mechanisms, the ways of performing tokenization, generating the query, key, and value matrices, and integrating attention in DRL could also influence sample efficiency.

---

### Official Review · Reviewer_SkMU · 2024-11-03

**Soundness:** 1
**Presentation:** 2
**Contribution:** 1
**Rating:** 1
**Confidence:** 5

**Summary:**

This paper claims to investigate the sample efficiency of self-attention mechanisms in image-based reinforcement learning. The paper mentions that they compare different types of scaled dot-product attention. However, the paper suffers from multiple major flaws including the model architecture as well as the choice of the self-attention operations used for experiments.

**Strengths:**

- The paper attempts to analyze different types of self-attention layers. The motivation is good.

**Weaknesses:**

- The paper uses a self-attention layer within a couple of CNN layers as their model architecture. This is a very shallow unusual architecture for any visual representation learning. In Transformers, it is a very common practice to have a series of MHSA layers and MLP layers interleaved. At least use ViT? The current architecture used for the investigation is extremely limited and no concrete conclusions can be made using them.

- The paper mentions that they investigate different self-attention. However, the selection of the attention mechanism to compare seems highly arbitrary. The operations selected in this paper include Spatial-wise-Attention, Channel-wise-Row(/Column)-Attention, and so on, but I have not seen any major state-of-the-art models using these arbitrary attention mechanisms in its architectures. Are these being used in any of today’s large vision-language models? What’s the point of comparing something that’s not being picked up by anyone? It would have been much more meaningful to compare different types of Transformer components that’s actually being used in practice, such as linear attention mechanisms like Performer and sequential models like Mamba2, which are much more of interest to the general audience.

- The observations from the experiments seem inconclusive. It is very difficult to conclude from these limited experiments which do not show any major trend.

**Questions:**

- It will be great if the authors can provide more justifications on their choice of the model architecture and the selection of the attention operations.

- What do you mean by this sentence in line 318? " To visualize the effect of self-attention modules in the feature space, we extract the attended feature maps [math symbol] which are the element-wise sum of the attention maps [math symbol] and the feature maps [math symbol] of the first CNN layer." What is the "[math symbol]"? This seems totally out of context?

---

> ### Author Response · Authors · 2024-12-01
>
> Dear Reviewer SkMU,
>
> Thank you for your time, valuable feedback, and insightful comments! We have revised our paper and would like to address the weaknesses and the questions as follows:
>
>
> **Responses to the weaknesses**
>
> >**W1**: The current architecture used for the investigation is extremely limited.
>
> We thank the reviewer's proposal of using larger and deeper transformer-based architectures such as ViT for this research topic and highlight the main motivations and justifications behind our choice of model architecture.
> - From the RL perspective, the model architecture used in our work is based on the PPO algorithm [1]. It is widely used in the RL community and can be regarded as the de facto policy optimization algorithm due to its data efficiency, reliable performance, and scalability. We briefly mentioned the rationale for choosing the PPO as the baseline agent in **Section 3.1**.
> - From the representation learning perspective, since the input of the models comprises four 84x84 low-dimensional Atari game frames in grayscale, we believe that 2 CNN layers (with 16 and 32 kernels respectively) should be sufficient to capture the state information for most of the games tested in this work.
> - The main objective of this work is to investigate the **direct** impact of the scaled dot-product attention on the sample efficiency of PPO from a dimensional perspective. Therefore, we prefer to adopt the simplest yet sufficient model architecture for such an investigation. The Transformer encoder contains a stack of MHSA layers and MLP layers with residual connections and layer normalizations which complicates our study on the self-attention layer alone.
> - Specifically in ViT, the image patches are projected into 1D vectors to form the $Q$, $K$, and $V$, and the dimension where the scaled dot-product attention can be applied is fixed which prohibits the study on the benefits of applying self-attention over different dimensions of the tokens.
>
> > **W2**: However, the selection of the attention mechanism to compare seems highly arbitrary.
>
> We acknowledge that the designs of the self-attention modules proposed in our work have certain limitations and highlight the main motivations of our design principles.
> - Our work is motivated by [2] where integrating self-attention between 2 CNN layers showed significant improvement in sample-efficiency in PPO based on the selected games.
> - In contrast to [2], we are interested in exploring the benefits of applying self-attention over different dimensions of the input tokens, i.e., the feature maps. Since the feature maps have 3 dimensions and the dot product is limited to 2 dimensions, we systematically devised the self-attention modules based on the combinations of the available dimensions.
> - We believe that the dimensional perspective of self-attention operation is worth investigating especially when the input tokens have more than 2 dimensions.
>
> >**W3**: The observations from the experiments seem inconclusive.
>
> We acknowledge that the contributions of this paper can be further improved by providing more conclusive statements based on the observations and offering more guidance to the community.
>
> **Responses to the questions**
>
> >**Q1**: It will be great if the authors can provide more justifications on their choice of the model architecture and the selection of the attention operations.
>
> Please refer to the responses to the weaknesses **W1** and **W2**.
>
> >**Q2**: What do you mean by this sentence in line 318?
>
> We are sincerely sorry for the typo. The math symbols have been corrected in the revised version of the paper near **line 414**.
>
> **References**
>
> [1] Proximal policy optimization algorithms (Schulman et al., 2017)
>
> [2] Reinforcement learning with attention that works: A self-supervised approach (Manchin et al., 2019)

---

> > ### Comment · Reviewer_SkMU · 2024-12-01
> >
> > 1. We disagree with the following statement.
> >
> > > From the RL perspective, the model architecture used in our work is based on the PPO algorithm [1]. It is widely used in the RL community and can be regarded as the de facto policy optimization algorithm due to its data efficiency, reliable performance, and scalability.
> >
> > We believe this statement is outdated. Please check this paper as an example [A1]. It uses PPO on a 7B Vision-Language Model. Another example: this paper [A2] uses PPO with Transformer-XL. [A1] https://arxiv.org/abs/2405.10292v2 [A2] https://arxiv.org/abs/2309.17207
> >
> > 2. The authors argue that 2 CNN layers are sufficient to capture the state information for most of the Atari games. This means that the proposed investigation is restricted to Atari-like game environment, and it will not transfer to any other environment with higher resolutions and more complicated scenes.
> >
> > 3. The authors also argue that they wanted to focus on the self-attention layers by removing all the other layers, but the objective of these self-attention layers is to be used within a large architecture, from the beginning. We question how meaningful it will be to just investigate the layers while not being part of any major architecture. There's no guarantee that the observations in this paper will scales to any state-of-the-art models or larger models like RT-1, RT-2, OpenVLA, and so on.
> >
> > 4. If the proposed work cannot be applied to ViT, that means the observation in this paper might be meaningless to most of the major visual representation architectures. Almost all state-of-the-art visual representations including CLIP, SigLIP, and DinoV2 rely on ViT.
> >
> > Thanks to the authors' feedback, we believe there is no misunderstanding and the pros and the cons of the paper is clear. I will increase my confidence to 5 accordingly while keeping the rating.

---

### Note · Authors · 2024-12-03

**Comment:**

We sincerely thank all the reviewers for their insightful and constructive feedback. We acknowledge the need for further efforts to enhance the conclusiveness and impact of our research, ensuring it provides greater value to the community. Once again, we deeply appreciate your time, dedication, and thoughtful review of our work.

**Withdrawal Confirmation:**

I have read and agree with the venue's withdrawal policy on behalf of myself and my co-authors.